# Flavor Quality Analysis of Ten *Actinidia arguta* Fruits Based on High-Performance Liquid Chromatography and Headspace Gas Chromatography–Ion Mobility Spectrometry

**DOI:** 10.3390/molecules28227559

**Published:** 2023-11-13

**Authors:** Jinli Wen, Yue Wang, Yanli He, Nan Shu, Weiyu Cao, Yining Sun, Pengqiang Yuan, Bowei Sun, Yiping Yan, Hongyan Qin, Shutian Fan, Wenpeng Lu

**Affiliations:** 1Institute of Special Animal and Plant Sciences, Chinese Academy of Agricultural Sciences, Changchun 130112, China; 2College of Food Science and Engineering, Jilin Agricultural University, Changchun 130018, China

**Keywords:** *Actinidia arguta*, HPLC, HS-GC-IMS, flavor, organic acids, volatile compounds

## Abstract

*Actinidia arguta* is a fruit crop with high nutritional and economic value. However, its flavor quality depends on various factors, such as variety, environment, and post-harvest handling. We analyzed the composition of total soluble sugars, titratable acids, organic acids, and flavor substances in the fruits of ten *A. arguta* varieties. The total soluble sugar content ranged from 4.22 g/L to 12.99 g/L, the titratable acid content ranged from 52.55 g/L to 89.9 g/L, and the sugar–acid ratio ranged from 5.39 to 14.17 at the soft ripe stage. High-performance liquid chromatography (HPLC) showed that citric, quinic, and malic acids were the main organic acids in the *A. arguta* fruits. Headspace gas chromatography–ion mobility spectrometry (HS-GC-IMS) detected 81 volatile compounds in 10 *A. arguta* varieties, including 24 esters, 17 alcohols, 23 aldehydes, 7 ketones, 5 terpenes, 2 acids, 1 Pyrazine, 1 furan, and 1 benzene. Esters and aldehydes had the highest relative content of total volatile compounds. An orthogonal partial least squares discriminant analysis (OPLS-DA) based on the odor activity value (OAV) revealed that myrcene, benzaldehyde, methyl isobutyrate, α-phellandrene, 3-methyl butanal, valeraldehyde, ethyl butyrate, acetoin, (E)-2-octenal, hexyl propanoate, terpinolene, 1-penten-3-one, and methyl butyrate were the main contributors to the differences in the aroma profiles of the fruits of different *A. arguta* varieties. Ten *A. arguta* varieties have different flavors. This study can clarify the differences between varieties and provide a reference for the evaluation of *A. arguta* fruit flavor, variety improvement and new variety selection.

## 1. Introduction

*Actinidia arguta* is a dioecious deciduous vine belonging to the genus Actinidia in the family Actinidiaceae [1]. It originates from China and is widely cultivated in various regions, such as Russia, Japan, Korea, the United States, and New Zealand [2]. Its fruit has a tart, sweet, and refreshing taste and is rich in vitamins, amino acids, proteins, minerals, and other nutrients [3,4]. It also contains bioactive compounds such as flavonoids, polysaccharides, and volatile oils [5,6]. It has various health benefits such as antioxidant, antitumor, and antihyperglycemic effects and is known as the “fruit of health” [7,8,9,10]. The Institute of Special Animal and Plant Sciences of the Chinese Academy of Agricultural Sciences has been collecting and breeding *A. arguta* varieties since the 1960 s and has developed ‘Kuilv’, ‘Fenglv’, ‘Jialv’, ‘Wanlv’, ‘Xinlv’, ‘Pinglv’, ‘Lvbao’, ‘Cuiyu’ and ‘Tianxinbao’ among others to promote the *A. arguta* industry [11].

Flavor is one of the key indicators of fruit quality, and it consists of acidity, sweetness, and aroma [12]. Flavor studies have been extensively studied in many fruit crops, such as grape [13], kiwifruit [12,14], and cherry [15]. Organic acids are important components of fruit flavor and functionality, affecting biological activity and microbial growth. Citric, malic, and tartaric acids are the most common in fruits, but the types and contents of organic acids vary widely among species and varieties [16]. *A. arguta* mainly contains citric and quinic acids, followed by malic, oxalic, mangiferin, and succinic acids [17]. Aroma compounds also contribute to the sensory quality of fruits [18]. A variety of aroma compounds, including esters, aldehydes, alcohols, and terpenes, have been detected in *A. arguta* so far. The expression of aroma is the result of the interaction of various aroma compounds, but generally, the compounds with higher content or distinctive aroma characteristics can largely determine the special aroma attributes of the fruit [19,20,21,22,23].

Nowadays, the main methods for the determination of organic acids in fruits and fruits include spectrophotometry, gas chromatography (GC), gas chromatography-mass spectrometry (GC-MS), high-performance liquid chromatography (HPLC), ion chromatography (IC), and liquid chromatography-mass spectrometry (LC-MS) [24,25]. Among them, HPLC has the advantages of good separation, high sensitivity, good accuracy, and a wide range of applications [25]. It has become an important separation and analytical technique in various fields such as chemistry, medicine, industry, agronomy, forensic science, and others [24,26,27].

Various methods have been used to detect volatile compounds in food, such as gas chromatography-olfactometry-mass spectrometry (GC-O-MS), gas chromatography-mass spectrometry (GC-MS), headspace solid-phase microextraction, and gas chromatography-ion mobility spectrometry (GC-IMS) [28]. Among them, GC-MS is the most widely used method for aroma analysis, but it has some drawbacks such as complex sample pre-processing and potential distortion [29,30]. GC-IMS is a novel method based on the differences in the migration speed of gas-phase ions in an electric field, which can achieve high separation and low detection limit, and has been widely applied in food volatile analysis [31,32,33]. From Table 1, it can be seen that the application of GC-IMS in *A. arguta* aroma analysis is still rare and needs further exploration.

In this study, we determined the total soluble sugar, titratable acid, organic acid, and aroma content of ten *A. arguta* varieties harvested in 2022, in Zuojia Town, Jilin City, Jilin Province, China, and established the fingerprints of volatile compounds in different varieties of *A. arguta* fruits. We also screened the key volatile compounds affecting the aroma of *A. arguta* fruits by combining OAV (odor activity value) analysis and VIP (variable importance in projection) analysis. This study aimed to investigate the differences between different *A. arguta* varieties, to understand the chemical basis of their flavor characteristics, and to provide a reference for the evaluation, improvement, and breeding of *A. arguta* varieties.

## 2. Results and Discussion

### 2.1. Analysis of Sugar–Acid Content of A. arguta Fruits of Different Varieties

The sugar–acid ratio can be derived by calculating the ratio of total soluble sugar content to titratable acid content, as a comprehensive index, it can reflect the sweet and sour taste of the fruit, which largely affects the flavor performance [12,39]. The total soluble solid, titratable acid content and sugar–acid ratio of each variety of *A. arguta* are shown in Figure 1, with the highest total acid content of ‘Fenglv’ (12.99 g/L) and the lowest of ‘Pinglv’ (4.22 g/L); the highest total sugar content of ‘Longcheng No.2’ (89.9 g/L), and the lowest was ‘Cuiyu’ (52.55 g/L). The largest sugar–acid ratio among the 10 varieties was ‘Pinglv’ (14.17), followed by ‘Tianxinbao’ (12.19) and the lowest was ‘Cuiyu’ (5.39).

### 2.2. Analysis of Organic Acids in A. arguta Fruit of Different Varieties

The type and content of organic acids affect the acidity, texture, and flavor of *A. arguta* fruits. Different tree species have different organic acids in their fruits, resulting in different flavors. For example, kiwifruit and summer orange contain malic acid, quinic acid, and citric acid; grape has malic acid and tartaric acid; pear and apple have mainly malic acid [40,41,42,43,44,45]. Table 2 shows the organic acid fractions and contents of different *A. arguta* varieties. Oxalic acid, quinic acid, malic acid, mangiferin, lactic acid, and citric acid were detected in *A. arguta* fruits, except for lactic acid in ‘Cuiyu’. Citric acid, quinic acid, and malic acid were the dominant organic acids in the ten varieties, accounting for more than 90% of the total acid content, which is consistent with previous studies [17,34,46]. Based on the organic acid components, the fruits can be categorized into citric acid dominant and quinic acid dominant groups [34]. ‘Kuilv’, ‘Fenglv’, ‘Wanlv’, ‘Xinlv’, ‘Pinglv’, ‘Lvbao’, ‘Cuiyu’, and ‘Longcheng No.2’ were citric acid dominant, while ‘Jialv’ and ‘Tianxinbao’ were quinic acid dominant. The differences in organic acid content were reflected in the titratable acidity level. ‘Fenglv’ had the highest titratable acidity level of 12.99 g/L: oxalic acid, quinic acid, malic acid, and citric acid all had the highest levels of 0.4 g/L, 8.48 g/L, 4.4 g/L, and 11.58 g/L, respectively, which were significantly higher than those of the other nine varieties (*p* < 0.05). The malic acid in fruits inhibits bacterial damage to the pulp, which facilitates fruit preservation [43,47]. ‘Lvbao’ had the highest mangiferin content of 0.1 g/L, which was significantly higher than the other nine varieties (*p* < 0.05). ‘Tianxinbao’ had the highest lactic acid content of 0.83 g/L, which was significantly higher than the other nine varieties (*p* < 0.05).

The heat map analysis can better reflect the characteristics of organic acids in different *A. arguta* samples. According to the heat map analysis of each variety (Figure 2), ten samples can be classified into three categories. The first category is ‘Fenglv’; the second category is ‘Kuilv’, ‘Longcheng No.2’, ‘Lvbao’, ‘Xinlv’, and ‘Wanlv’; the third category is ‘Tianxinbao’, ‘Cuiyu’, ‘Jialv’, and ‘Pinglv’. The content of organic acids varied greatly among the samples. The highest total amount of organic acids (oxalic, quinic, malic, lactic, and citric acids were all higher) was found in ‘Fenglv’; the total amount of organic acids were higher in the second category; and the total amount of organic acids was lower in the third category.

### 2.3. Analysis of Volatile Flavor Substances in Different Varieties of A. arguta Fruits

#### 2.3.1. Two-Dimensional Top-View Spectrogram Analysis

Figure 3a shows the ion mobility spectra of the volatile substances in ten varieties of *A. arguta* fruits. The horizontal coordinate represents the relative drift time/RIP (no unit), and the vertical coordinate represents the GC retention time (s). The color depth indicates the content of the volatile substances, with darker colors indicating higher concentrations. The results show that GC can separate the volatile components in different varieties of *A. arguta* fruits within 30 min and that there are differences in the volatile flavor substances among them. This suggests that the proportion composition of volatiles is one of the key material bases for the flavor diversity of *A. arguta* fruits.

#### 2.3.2. Difference Spectrum Analysis

To better compare the differences in the volatile components of different varieties of *A. arguta* fruits, we used the ‘Kuilv’ variety as a reference and subtracted its signal peaks from the rest of the spectra to obtain the difference spectra. The blue area indicates that the content of the substance in this sample is lower than that in ‘Kuilv’, and the red area indicates that the content of the substance in this sample is higher than that in ‘Kuilv’. Again, the darker the color, the greater the difference. Figure 3b shows the difference spectrum, from which it can be seen that ‘Lvbao’ contains richer volatile flavor substances, followed by ‘Tianxinbao’. The concentration of volatile substances in ‘Kuilv’ is similar to that in ‘Fenglv’. ‘Jialv’, ‘Wanlv’, ‘Xinlv’, and ‘Pinglv’ are more similar to each other. ‘Cuiyu’ has the lowest volatile flavor substance concentration.

#### 2.3.3. Qualitative Analysis of Volatile Compounds in Different Varieties of *A. arguta* Fruits

We calculated the retention index of each volatile compound in *A. arguta* fruits using C4~C9 ortho-ketone as the external standard reference, based on the GC retention time and IMS migration time of the volatile compounds. We characterized the volatile compounds in *A. arguta* fruits by using the built-in NIST 2020 database and the IMS database of HS-GC-IMS. Appendix A shows that the volatile components characterized in 10 different varieties of *A. arguta* contained a total of 81 monomer and dimer substances, including 24 esters, 17 alcohols, 23 aldehydes, 7 ketones, 5 terpenes, 2 acids, 1 Pyrazine, 1 furan, and 1 benzene. These substances collectively constitute the characteristic flavor of *A. arguta* fruits.

#### 2.3.4. Gallery Plot Fingerprint Analysis of Volatiles in *A. arguta* Fruits

We analyzed the differences in volatile flavor compounds in different varieties of *A. arguta* by plotting the fingerprints of volatile flavor compounds with the Gallery Plot plug-in (Figure 4), based on three replicates of each sample. The color depth indicates the intensity and content of the peaks, with darker colors indicating higher values. The fingerprints show the composition and differences in volatile flavor compounds in different *A. arguta* fruit samples. Among them, ‘Kuilv’ had higher contents of butanal M, valeraldehyde M, ethyl acetate, terpinolene, and β-pinene; ‘Fenglv’ had higher contents of (E)-2-heptenal D, (E)-2-hexenal D, (E)-2-pentenal D, (E)-2-pentenal M, 1-hexanal D, heptanal M, 1-penten-3-ol, 3-methyl-2-butanol, 1-penten-3-one D, acetone, methyl isobutyrate, etc.; ‘Jialv’ had higher contents of (E)-2-hexenal M, 1-hexanal D, 1-hexanal M, valeraldehyde M, 1-penten-3-one M, etc.; ‘Wanlv’ had higher contents of 1-nonanal, 3-octanone, ethyl formate M, α-Phellandrene, and myrcene; ‘Xinlv’ had higher contents of 1-hexanal D, 3-Methyl butanal, 1-octen-3-ol, 3-methyl-2-butanol, ethyl butyrate M, 2,5-dimethylfuran and other substances; ‘Pinglv’ had higher contents of (E)-2-hexenal M, hexanal M, valeraldehyde M and 1-penten-3-one M; ‘Lvbao’ had higher contents of (Z)-4-heptenal D, (Z)-4-heptenal M, butanal D, valeraldehyde D, isobutanol D, isobutanol M, acetic acid M, propyl acetate, ethyl butyrate D, butyl isovalerate D, butyl isovalerate M, butyl acetate D, butyl acetate M, hexyl acetate, isopentyl acetate D, isobutyl acetate M, isobutyl acetate, pentyl acetate D, pentyl acetate M, and propyl propionate; ‘Cuiyu’ had higher contents of diethyl acetal, isoamyl alcohol M, 1-pentanol, 1-propanol, 1-butanol, acetoin D, acetoin D, and 2-pentanone; ‘Tianxinbao’ had higher contents of (E)-2-heptenal M, isopentanol D, 1-penten-3-ol, 2-heptanol, 2-pentanone, butyl isovalerate M, methyl isobutyrate, methyl acetate; and ‘Longcheng No.2’ had higher contents of (E)-2-octenal D, (E)-2-octenal M, 1-hexanol D, 1-hexanol M.

### 2.4. Content Analysis of Volatile Compounds in A. arguta Fruits

The aroma and flavor of *A. arguta* fruits depend on the content and proportion of aroma components, especially the variation of the component composition [13], which differs significantly among different varieties. Figure 5 shows that aldehydes were the main contributing components in ‘Kuilv’, ‘Fenglv’, ‘Jialv’, ‘Wanlv’, ‘Xinlv’, ‘Pinglv’, ‘Cuiyu’, ‘Tianxinbao’ and ‘Longcheng No.2’; and esters were the main contributing components in ‘Lvbao’. The ten varieties of *A. arguta* had the same types of volatile compounds detected in the fruit, but the contents varied greatly. The highest content of volatile compounds was detected in ‘Lvbao’ (21,733.59 μg/kg), followed by ‘Tianxinbao’ (18,322.04 μg/kg), ‘Fenglv’ (18,106.55 μg/kg), ‘Xinlv’ (17,256.37 μg/kg), ‘Longcheng No.2’ (16,340.77 μg/kg), ‘Kuilv’ (16,106.28 μg/kg), ‘Jialv’ (16,070.06 μg/kg), and ‘Wanlv’ (15,531.64 μg/kg).

#### 2.4.1. Esters

Esters play an important role in the formation of the aroma profile of *A. arguta* [22,23,38]. Appendix A shows that esters are the most abundant types of volatile compounds detected in each variety, and some important esters, such as ethyl acetate, butyl acetate, methyl isobutyrate, and isopentyl acetate, can give *A. arguta* fruits a strong fruity and floral aroma. The highest content of esters was found in ‘Lvbao’ (11,145 μg/kg), followed by ‘Tianxinbao’ (4348.81 μg/kg), ‘Cuiyu’ (2986.71 μg/kg), ‘Longcheng No.2’ (2641.47 μg/kg), ‘Xinlv’ (1716.49 μg/kg), ‘Fenglv’ (1696.37 μg/kg), ‘Pinglv’ (1604.62 μg/kg), ‘Kuilv’ (1464.64 μg/kg), ‘Wanlv’ (1450.78 μg/kg), and ‘Jialv’ (1395.84 μg/kg).

#### 2.4.2. Aldehydes

Aldehydes accounted for a large proportion of the total volatile compounds in 10 varieties of *A. arguta*, ranging from 14.46% to 62%. The highest content of aldehydes was found in ‘Fenglv’ (12,038.21 μg/kg), followed by ‘Xinlv’ (11,200.15 μg/kg), ‘Jialv’ (11,154.69 μg/kg), ‘Kuilv’ (10,135.59 μg/kg), ‘Wanlv’ (9637.65 μg/kg), ‘Pinglv’ (9369.09 μg/kg), ‘Tianxinbao’ (6955.3 μg/kg), ‘Longcheng No.2’ (26,845.95 μg/kg), ‘Lvbao’ (3644.42 μg/kg), and ‘Cuiyu’ (3593.72 μg/kg). Aldehydes mainly contributed to the green grass and vegetable aroma of *A. arguta* [48,49], which indicated that ‘Fenglv’ had a stronger grassy flavor.

#### 2.4.3. Alcohols

The content of alcohols in each variety varied from 1607.69 μg/kg to 3639.13 μg/kg, with the highest content found in ‘Tianxinbao’ (3639.13 μg/kg), followed by ‘Longcheng No.2’ (3538.89 μg/kg), ‘Lvbao’ (3486.54 μg/kg), ‘Cuiyu’ (3452.6 μg/kg), ‘Pinglv’ (1937.99 μg/kg), ‘Kuilv’ (1947.74 μg/kg), ‘Fenglv’ (1945.78 μg/kg), ‘Xinlv’ (1869.8 μg/kg), ‘Wanlv’ (1721.37 μg/kg), and ‘Jialv’ (1607.69 μg/kg).

#### 2.4.4. Ketones

The content of ketones ranged from 1281.44 μg/kg to 2499.92 μg/kg, accounting for 7.54–18.23% of the total volatile components, with the highest content in ‘Cuiyu’ and the lowest in ‘Kuilv’. The ketones detected in the fruits of the 10 *A. arguta* varieties were mainly 1-penten-3-one, 2-pentanone, and acetone, which had a green, pungent, buttery flavor [29,30].

#### 2.4.5. Others

The total concentration of terpenoids, acids, furans, pyrazines, and benzenes was low, accounting for only 0.6–4.58%, 1.06–6.21%, 0.42–0.78%, 0.03–0.23%, and 0.03–0.31% of the samples of each variety, respectively.

### 2.5. PCA Analysis of A. arguta Fruit Aroma Substances

To analyze the differences between different *A. arguta* fruit samples more intuitively, we performed PCA analysis on the volatile compounds identified by HS-GC-IMS. The ten samples were well differentiated by their aroma characteristics and varieties. PC1 accounted for 42.9% of the variance, and PC2 accounted for 14.1%. The ten groups of samples showed a clear separation trend on the 2D graph, with no outliers. The samples of the same species of *A. arguta* fruit clustered well. The PCA results indicated significant differences in the overall aroma substances of the ten groups of samples. Figure 6 shows that ‘Kuilv’, ‘Fenglv’, ‘Jialv’, ‘Wanlv’, ‘Xinlv’ and ‘Pinglv’ were closer to each other, ‘Longcheng No.2’ and ‘Tianxinbao’ were closer to each other, while ‘Lvbao’ was far from the other nine varieties, suggesting significant differences in the aroma characteristics of different samples. The HS-GC-IMS technique can distinguish different varieties of *A. arguta* fruits and explore their characteristics. The method is fast and non-destructive.

### 2.6. OAV Analysis of Aroma Components of Different Varieties of A. arguta Fruit

The intensity of *A. arguta* fruit aroma depends on the concentration and threshold value of the aroma components. Only when the concentration is higher than the threshold value (OAV ≥ 1), the component is considered an active aroma component. We calculated that 25 volatile aroma substances with OAVs > 1 were detected in ten *A. arguta* fruits (Table 3). Among them, nine were aldehydes: (E)-2-hexenal, (E)-2-octenal, (Z)-4-heptenal, 1-hexanal, 1-nonanal, valeraldehyde, valeraldehyde, 3-Methyl butanal and benzaldehyde; nine were esters: ethyl butyrate, butyl isovalerate, butyl acetate, hexyl acetate, hexyl propanoate, isopentyl acetate, isobutyl acetate and methyl butyrate; four were terpenes: α-phellandrene, α-pinene, myrcene and terpinene; two were ketones: 1-penten-3-one and acetoin; and one was alcohol: 1-octen-3-ol. Although the OAV values of the key volatile compounds varied among the 10 *A. arguta* varieties, aldehydes generally had higher OAV values than other types of volatile compounds. Because of their lower aroma threshold, they contributed more to the aroma of *A. arguta* fruit even at lower concentrations. The OAVs of 1-octen-3-ol in all varieties ranged from 23.95 to 69.29, which significantly contributed to the *A. arguta* fruit aroma.

#### 2.6.1. Heat Map Analysis of Volatile Compounds with OAVs > 1

We used hierarchical analysis to cluster the volatile aroma substances with OAVs > 1 in ten *A. arguta* fruits and clarify the affinity of volatile flavor profiles among different samples. The heat map analysis (Figure 7a) showed that ‘Lvbao’ was clustered into one class alone, ‘Cuiyu’ and ‘Longcheng No.2’ clustered into another class, and ‘Kuilv’, ‘Fenglv’, ‘Jialv’, ‘Wanlv’, ‘Xinlv’, ‘Pinglv’, and ‘Tianxinbao’ clustered into a third group. The red color indicated that the volatile compound was highly expressed in the sample, and the blue color indicated that the volatile compound was less expressed in the sample. The content of volatile aroma substances with OAVs > 1 varied greatly among the samples. Among them, butyl acetate, isoamyl acetate, isobutyl acetate, (Z)-4-heptenal, and hexyl acetate were highly expressed in ‘Lvbao’; α-phellandrene and myrcene were highly expressed in ‘Wanlv’; hexyl acetate and (E)-2-octenal were highly expressed in ‘Longcheng No.2’; and acetoin was highly expressed in ‘Cuiyu’.

#### 2.6.2. Correlation Analysis of Volatile Compounds with OAVs > 1

The absolute range of correlation coefficients between substances was between 0.8 and 1.0 indicating very strong correlation, between 0.6 and 0.8 indicating strong correlation, between 0.4 and 0.6 indicating moderate correlation, between 0.2 and 0.4 indicating weak correlation, and between 0 and 0.2 indicating very weak or no correlation between substances [13]. In the correlation analysis in Figure 7b, the Pearson correlation coefficients for the red boxes were positively correlated. (E)-2-hexenal correlated very strongly with 1-hexanal, 1-penten-3-one, valeraldehyde, and strongly with 1-nonanal, 1-octen-3-ol, and methyl isobutyrate; (Z)-4-heptenal correlated very strongly with butyl isovalerate, butyl acetate, hexyl acetate, isobutyl acetate, and isopentyl acetate; hexanal correlated extremely with 1-nonanal, 1-penten-3-one, and very strongly with 1-octen-3-ol, methyl butyrate, valeraldehyde; 1-nonanal correlated very strongly with 1-octen-3-ol, strongly with 1-penten-3-one, 3-methyl butanal; 1-octen-3-ol correlated very strongly with 3-methyl butanal, strongly with 1-penten-3-one; 1-penten-3-one correlated strongly with 3-methyl butanal, methyl isobutyrate, valeraldehyde; α-phellandrene correlated strongly with α-pinene, myrcene, terpinolene; α-pinene correlated very strongly with terpinolene; butyl isovalerate correlated very strongly with butyl acetate, hexyl acetate, isopentyl acetate, and isobutyl acetate; butyl acetate correlated very strongly with hexyl acetate, isopentyl acetate, and isobutyl acetate; hexyl acetate correlated very strongly with isopentyl acetate and isobutyl acetate; and Pearson correlation coefficients for the blue box were negatively correlated. Acetoin was very significantly negatively correlated with (E)-2-hexenal, 1-hexanal, and 1-penten-3-one; and (E)-2-octenal was very significantly negatively correlated with pentanal.

#### 2.6.3. OPLS-DA Analysis of Volatile Compounds with OAVs > 1

OPLS-DA is a supervised discriminant analysis statistical method that can model the relationship between substance expression and samples to achieve the prediction of sample categories [50]. We performed partial least squares discriminant analysis on the volatile fractions of different *A. arguta* fruits with OAV values greater than 1 and selected the differential volatiles with VIP > 1 and *p* < 0.05 as the characteristic volatiles. We screened 13 characteristic volatile substances that contributed significantly to the classification model: four aldehydes (benzaldehyde, 3-methyl butanal, valeraldehyde, (E)-2-octenal), four esters (methyl isobutyrate, ethyl butyrate, hexyl acetate, methyl butanoate), three terpenes (myrcene, α-phellandrene, terpinolene), and two ketones (acetoin, 1-penten-3-one). These substances may be important for distinguishing different varieties of *A. arguta* (Figure 8a). Myrcene, which had the highest VIP value, was one of the most important markers for the differences in the aroma characteristics of *A. arguta* fruits from different varieties. Figure 8b shows the structure of the key volatile compounds.

## 3. Materials and Methods

### 3.1. Materials and Reagents

#### 3.1.1. Plant Materials

Ten *A. arguta* varieties were harvested during in September 2022 from the *Actinidia arguta* Resource Nursery of the Institute of Special Animal and Plant Sciences, Chinese Academy of Agricultural Sciences, Zuojia Town, Jilin City, Jilin Province. *A. arguta* fruits are shown in Appendix A. Sampling was performed by randomly selecting well-grown, medium-sized fruit trees in the resource nursery, choosing *A. arguta* with the same degree of exposure to light, the same size, and similar hardness and that was free of pests and diseases. The fruits were analyzed at the eating-ripe stage. Sample information is presented in Appendix A.

#### 3.1.2. Overview of the Sampling Sit

*Actinidia arguta* Resource Nursery was on a gentle slope in the mountains, using dark brown forest soil. After the soil thawed in spring, we dug a 40 cm diameter and 30 cm deep planting hole in the center of a cultivation ditch, where we planted ten different varieties of *A. arguta* (a perennial vine fruit plant). The seedlings’ roots spread evenly in the hole. We adopted a T-shaped frame cultivation method, with a row spacing of 3.5 m × 2.0 m and a male-to-female plant ratio of 8:1. We applied fertilizer 2–3 times and removed weeds 3–4 times annually. The sampling sites are shown in Appendix A.

#### 3.1.3. Reagents and Instruments

Test Reagents: All chemicals used were of analytical grade or better. Sodium hydroxide, sulfuric acid (Beijing Chemical Factory, Beijing, China); anthrone, anhydrous ethanol, phenolphthalein (Sinopharm Chemical Reagents Co., Ltd., Shanghai, China); anhydrous glucose (Xilong Chemical Co., Ltd., Guangzhou, China); oxalic acid, malic acid, shikimic acid, citric acid, quinic acid (Shanghai Yuanye Biotechnology Co., Ltd., Shanghai, China); methanol (TEDIA Reagents, Fairfield, OH, USA); 4-methyl-2-pentanol (Shanghai Aladdin Biochemical Technology Co., Ltd., Shanghai, China); lactic acid (Tianjin Institute of Fine Chemical Industry, Tianjin, China).

Instruments: Electronic balance (Cany Precision Instruments Co., Ltd., Shanghai, China); Cary 60 UV-Vis spectrophotometer, high-performance liquid chromatograph (Agilent Technologies Co., Ltd., Waldbronn, Germany); KQ-300 E Ultrasonic Cleaner (Kunshan Ultrasonic Instrument Co., Ltd., Kunshan, China); XH-D vortex mixer (Wuxi Woxin Instrument Co., Ltd., Wuxi, China); Allegra 64 R High-speed freezing centrifuge (Beckman Coulter, Inc., Carlsbad, CA, USA); FlavourSpec^®^ flavor analyzer (G.A.S.).

### 3.2. Experimental Methods

#### 3.2.1. Determination of Sugar and Acid Content

The total soluble sugar content of *A. arguta* juice was determined using anthrone and sulfuric acid colorimetry; the titratable acid content of *A. arguta* juice was determined by the NaOH neutralization titration method [4,13]. Sugar–acid ratio = total soluble sugar content/titratable acid content [12]. It is a comprehensive index that reflects the balance of sweetness and sourness in the fruit, which largely influences its flavor quality [39].

#### 3.2.2. Detection of Organic Acids Content

The organic acid content of *A. arguta* was determined by high-performance liquid chromatography (HPLC) [51].

Standard preparation: The following organic acids were weighed: 0.01030 g of oxalic acid, 0.0108 g of quinic acid, 0.0103 g of malic acid, 0.0100 g of shikimic acid, and 0.0103 g of citric acid. Also, 0.1 mL of lactic acid was measured. The concentration of each acid was adjusted to 10 mL with the mobile phase, resulting in the following concentrations: 1.03 g/L for oxalic, malic, and citric acids; 1.08 g/L for quinic acid; 1.00 g/L for shikimic acid; and 10.61 g/L for lactic acid. Gradient dilution was performed to obtain the regression equations and correlation coefficients of the peak areas (x) versus the mass concentrations (Y) of the organic acids. The standard curves of the organic acids were plotted and are shown in Appendix A.

Sample pretreatment: The sample was diluted by a factor of two with the mobile phase, filtered through a 0.22 μm microporous filter membrane, and reserved for injection.

Chromatographic conditions: A C18-XT column (4.6 mm × 250 mm, 5 μL) was used at a column temperature of 25 °C and a detection wavelength of 210 nm. The mobile phase was aqueous phosphoric acid (pH 2.3) with a flow rate of 0.3 mL/min and an injection volume of 10 μL.

#### 3.2.3. Detection of Volatile Compounds

The volatile substances of *A. arguta* fruit were determined by headspace gas chromatography–ion mobility spectrometry (HS-GC-IMS), referring to the previously published literature [28]. Sample treatment: Take 3 g of *A. arguta* fruit homogenate in a 20 mL headspace vial, and add 20 μL of 10 ppm internal standard (4-methyl-2-pentanol). The internal standard concentration was 198 ppb, the signal peak volume was 478.01, and the intensity of each signal was approximately 0.414 ppb. Substances were qualitatively analyzed in terms of C4–C9 ketones to calculate retention indices, using the software’s built-in NIST and IMS databases. The analytical conditions and gas chromatographic conditions are shown in Appendix A. The quantitative calculation of volatile compounds is as follows:$$C_{i}=\frac{C_{is}\times A_{i}}{A_{is}}$$

C_i_ is the calculated mass concentration of any volatile component, in µg/kg, C_is_ is the mass concentration of the internal standard (4-Methyl-2-pentanol), in µg/kg, and A_i_/A_is_ is the volume ratio of any signal peak to the signal peak of the internal standard.

#### 3.2.4. Odor Activity Value (OAV) Calculation

The contribution of the overall aroma of *A. arguta* fruit fruit was evaluated using the odor activity value (OAV). The OAV was calculated using the formula [13,28,52].
$$OAV=\frac{Cx}{OTx}$$

OAV is the aroma activity value; Cx is the concentration of volatile compound x (μg/kg); and OTx is the aroma threshold of volatile compound x in water (μg/kg), which is mainly referred to in the *Compendium of Compound Aroma Thresholds* (Second Edition) and the aroma thresholds of compounds reported in the literature [48,49]. The OAV was calculated by dividing the concentration of volatile compounds by the odor threshold. Volatile compounds with OAV > 1 were considered aroma-active and played an important role in developing the aromatic properties of *A. arguta* fruit.

### 3.3. Data Processing and Statistical Analyses

Each measurement was performed in triplicate; the data are presented as mean ± standard deviation and differences between groups were deemed significant at *p* < 0.05. The data were analyzed using Excel 2016 and SPSS 27.0 for ANOVA and significance testing. Figure 1 was created using Origin pro2021. OPLS-DA and VIP value analysis were conducted using Simca14.1. Heat map analysis, PCA, and correlation analysis were performed using the omicshare tool. (https://www.omicshare.com/tools, accessed 8 August 2023).

## 4. Conclusions

*Actinidia arguta* is a fruit crop with high nutritional and economic value, but its flavor quality is affected by various factors. We analyzed the flavor components and their variation among different varieties of *Actinidia arguta*, and identified the key compounds that contribute to the unique aroma of this fruit: we measured the total soluble sugars, titratable acids, organic acids and volatile compounds in the fruits of ten *Actinidia arguta* varieties harvested in 2022 in Zuojia Town, Jilin City, China. We found that each variety had a distinctive flavor. ‘Lvbao’ and ‘Tianxinbao’ had the highest sugar to acid ratio and the highest total volatile compounds, indicating their superior fruit flavor. High-performance liquid chromatography (HPLC) was used to determine the content of organic acids. The results showed that citric acid, quinic acid and malic acid were the major organic acids, with ‘Fenglv’ having the highest total organic acid.

Headspace gas chromatography–ion mobility spectrometry (HS-GC-IMS) was used to determine the volatile compounds in different varieties of *Actinidia arguta* fruits, and a total of 81 volatile compounds were identified, including 24 esters, 17 alcohols, 23 aldehydes, 7 ketones, 5 terpenes, 2 acids, 1 pyrazine, 1 furan, and 1 benzene. Esters and aldehydes had the highest relative amounts. The fingerprints of volatile substances in *Actinidia arguta* fruits were established. The fingerprints could effectively distinguish different varieties and could be directly used for quality characterization and variety identification. An orthogonal partial least squares discriminant analysis (OPLS-DA) based on the odor activity value (OAV) revealed that myrcene, benzaldehyde, methyl isobutyrate, α-phellandrene, 3-methyl butanal, valeraldehyde, ethyl butyrate, acetoin, (E)-2-octenal, hexyl propanoate, terpinolene, 1-penten-3-one, and methyl butyrate were the main contributors to the differences in the aroma profiles of the fruits—these compounds can be used as markers for *Actinidia arguta* fruit flavor evaluation and improvement.

Headspace gas chromatography–ion mobility spectrometry (HS-GC-IMS) can show the commonalities and differences between the samples, which can complement the sensory evaluation and play a useful role in the assessment of the flavor quality of *Actinidia arguta* fruits. The results of this study can clarify the differences between varieties and provide a reference for the evaluation of arrowroot fruit flavor, variety improvement and new variety selection. However, the NIST database and the IMS database are not complete enough, which prevents some compounds isolated by GC from being characterized. Therefore, the gradual enrichment of the database is an important development direction for the detection of volatile compounds in the future. Meanwhile, it is necessary to establish a more detailed quality evaluation system for *Actinidia arguta* by combining appearance quality, nutritional quality, processing quality and flavor quality.

## Figures and Tables

**Figure 1 molecules-28-07559-f001:**
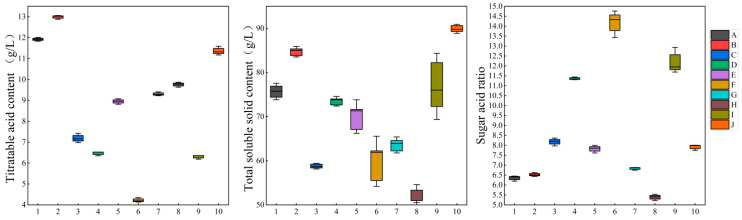
Contents of sugars and acid and sugar to acid ratio in different *Actinidia arguta* cultivars. Note: From A to J, they are ‘Kuilv’, ‘Fenglv’, ‘Jialv’, ‘Wanlv’, ‘Xinlv’, ‘Pinglv’, ‘Lvbao’, ‘Cuiyu’, ‘Tianxinbao’, and ‘Longcheng No.2’.

**Figure 2 molecules-28-07559-f002:**
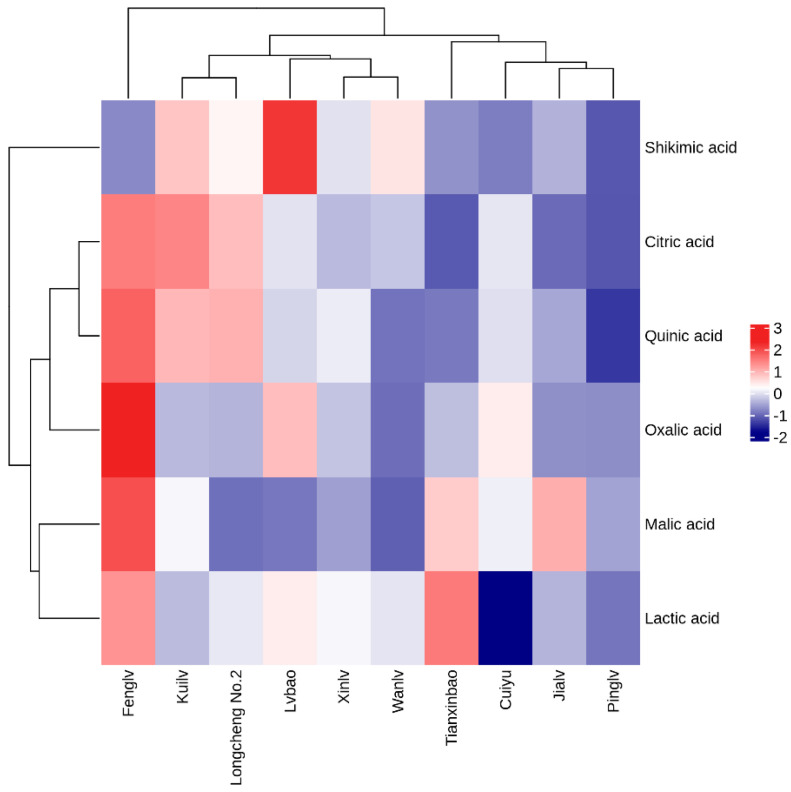
Heat map analysis of organic acids in fruits of different *A. arguta* varieties.

**Figure 3 molecules-28-07559-f003:**
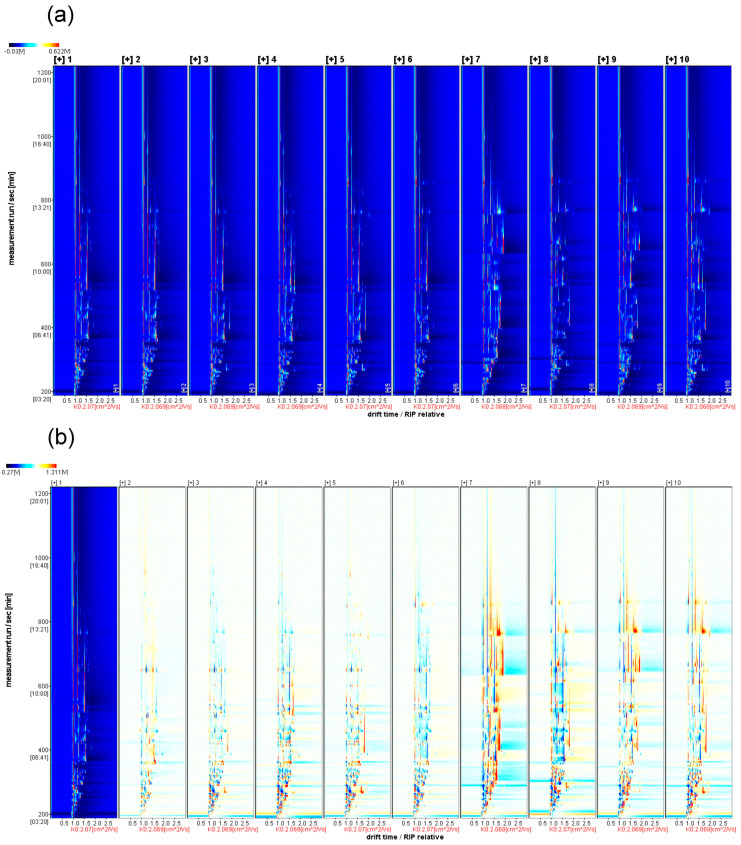
(**a**) The two-dimensional spectrum of different *A. arguta* fruit samples; (**b**) Note: HS-GC-IMS difference spectrum of different *A. arguta* fruit samples. From 1 to 10, they are ‘Kuilv’, ‘Fenglv’, ‘Jialv’, ‘Wanlv’, ‘Xinlv’, ‘Pinglv’, ‘Lvbao’, ‘Cuiyu’, ‘Tianxinbao’, and ‘Longcheng No.2’.

**Figure 4 molecules-28-07559-f004:**
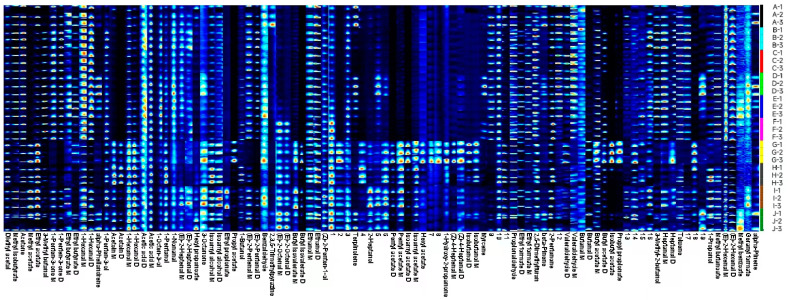
Fingerprints of volatile compounds in different *A. arguta* fruits. Note: From A to J, they are ‘Kuilv’, ‘Fenglv’, ‘Jialv’, ‘Wanlv’, ‘Xinlv’, ‘Pinglv’, ‘Lvbao’, ‘Cuiyu’, ‘Tianxinbao’, and ‘Longcheng No.2’.

**Figure 5 molecules-28-07559-f005:**
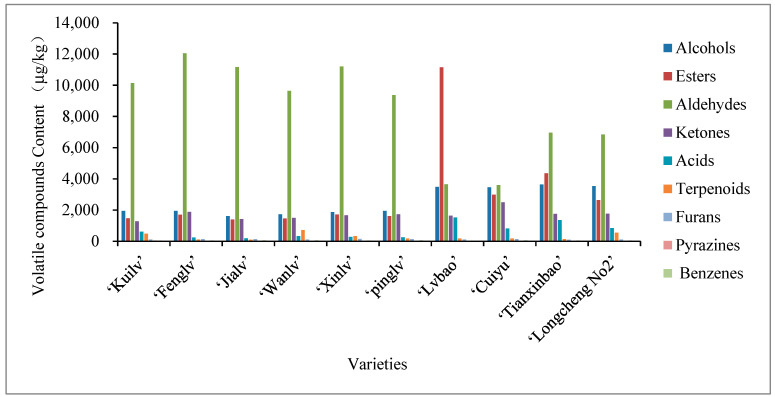
Histogram of volatile aromatic compounds in fruits of different *A. arguta* varieties.

**Figure 6 molecules-28-07559-f006:**
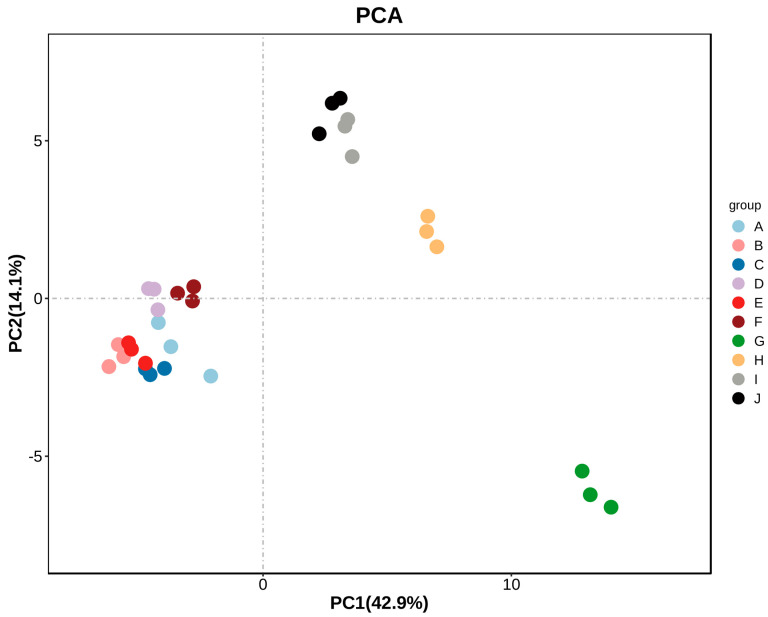
PCA analysis of volatile aroma compounds in fruits of different *A. arguta* varieties. Note: From A to J, they are ‘Kuilv’, ‘Fenglv’, ‘Jialv’, ‘Wanlv’, ‘Xinlv’, ‘Pinglv’, ‘Lvbao’, ‘Cuiyu’, ‘Tianxinbao’, and ‘Longcheng No.2’.

**Figure 7 molecules-28-07559-f007:**
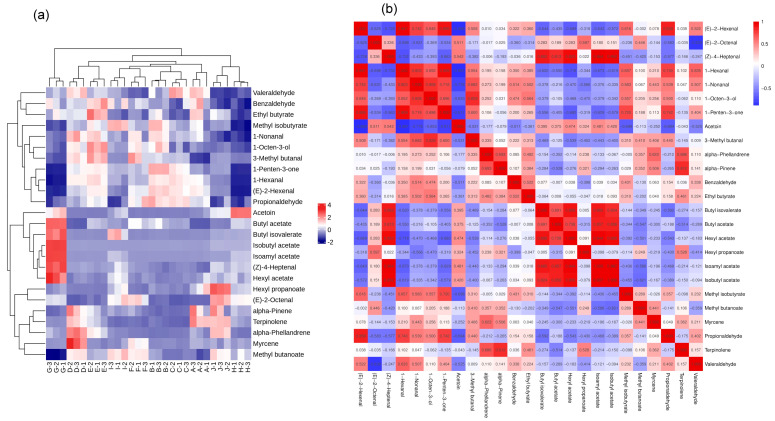
(**a**) Heat map analysis of volatile aroma compounds with OAVs > 1 in fruits of different *A. arguta* varieties. Note: From A to J, they are ‘Kuilv’, ‘Fenglv’, ‘Jialv’, ‘Wanlv’, ‘Xinlv’, ‘Pinglv’, ‘Lvbao’, ‘Cuiyu’, ‘Tianxinbao’, and ‘Longcheng No.2’. (**b**) Correlation analysis of volatile aroma compounds with OAVs > 1 in fruits of different *A. arguta* varieties.

**Figure 8 molecules-28-07559-f008:**
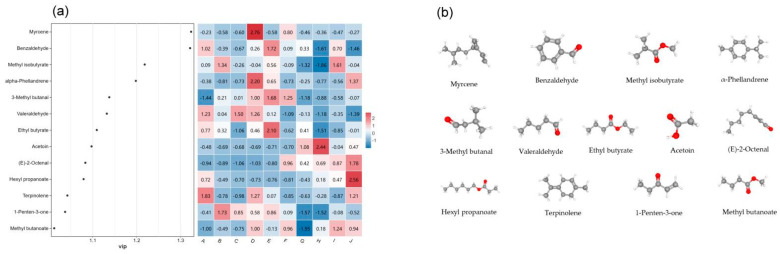
(**a**) Variable importance in projection(VIP) scores based on OPLS-DA of odor activity value(OAV). Note: From A to J, they are ‘Kuilv’, ‘Fenglv’, ‘Jialv’, ‘Wanlv’, ‘Xinlv’, ‘Pinglv’, ‘Lvbao’, ‘Cuiyu’, ‘Tianxinbao’, and ‘Longcheng No.2’. (**b**) Structure of key volatile compounds.

**Table 1 molecules-28-07559-t001:** Studies on organic acids and volatile substances in *A. arguta* fruit.

	Research Methods	Major Compounds	Reference
Organic acids	HPLC	Citric acid, quinic acid, ascorbic acid, and malic acid	[34]
	HPLC	citric acid, quinic acid	[35]
Volatile compounds	GC-O, GC-MS	Ethyl butanoate, Hexanoate, 2-Methylbutanoate, 2-Methylpropanoate, Hexanal and Hex-E2-enal	[7]
	GC-MS	Ethyl butanoate, Furaneol, 1-Penten-3-one, Pentanal, Hexanal, (E)-2-Hexenal, 1-Octen-3-ol, Linalool, Terpinen-4-ol, and α-terpineol	[21]
	GC-MS	1-Methyl-4-(1-methylethylidene)-cyclohexene, Butanoic acid ethyl ester, Ethanol, Hexanoic acid ethyl ester, Benzoic acid methyl ester, β-Myrcene,D-Limonene and β-Pinene	[22]
	GC-MS	Ethyl butyrate	[23]
	GC-IMS	Isoamyl acetate, 3-Methyl-1-butanol, 1-Hexanol, and Butanal	[35]
	GC-MS	Ethyl butanoate	[36]
	GC-MS	2,5-Dimethyl-4-hydroxy-3(2 H)-furanone (Furaneol), Benzyl alcohol, 3-Hydroxy-beta-damascone, Hexanal, and (Z)-3-Hexen-1-ol	[37]
	GC-MS	E-2-Hexenal	[38]

**Table 2 molecules-28-07559-t002:** Comparison of organic acids in fruit samples of different *A. arguta* varieties.

Variety	Oxalic Acid g/L	Quinic Acid g/L	Malic Acid g/L	Shikimic Acid g/L	Lactic Acid g/L	Citric Acid g/L
‘Kuilv’	0.09 ± 0.01 ^d^	7.01 ± 0.51 ^c^	2.48 ± 0.15 ^d^	0.06 ± 0.01 ^b^	0.37 ± 0.02 ^e^	11.34 ± 0.76 ^b^
‘Fenglv’	0.4 ± 0.05 ^a^	8.48 ± 0.33 ^a^	4.4 ± 0.23 ^a^	0.03 ± 0.01 ^ef^	0.77 ± 0.02 ^b^	11.58 ± 0.95 ^a^
‘Jialv’	0.05 ± 0.01 ^e^	4.43 ± 0.216 ^g^	3.42 ± 0.29 ^b^	0.03 ± 0.01 ^de^	0.36 ± 0.09 ^e^	3.65 ± 0.13 ^h^
‘Wanlv’	0.01 ± 0.01 ^f^	3.69 ± 0.11 ^i^	1.12 ± 0.15 ^j^	0.06 ± 0.01 ^bc^	0.46 ± 0.02 ^d^	6.04 ± 0.24 ^f^
‘Xinlv’	0.1 ± 0.01 ^d^	5.47 ± 0.15 ^d^	1.68 ± 0.16 ^g^	0.04 ± 0.01 ^cd^	0.5 ± 0.03 ^d^	5.74 ± 0.25 ^g^
‘Pinglv’	0.04 ± 0.01 ^e^	2.87 ± 0.21 ^j^	1.71 ± 0.08 ^f^	0.02 ± 0 ^f^	0.23 ± 0.06 ^f^	3.17 ± 0.16 ^j^
‘Lvbao’	0.23 ± 0.01 ^b^	5.12 ± 0.11 ^f^	1.31 ± 0.06 ^h^	0.1 ± 0.02 ^a^	0.56 ± 0.04 ^c^	6.79 ± 0.33 ^e^
‘Cuiyu’	0.17 ± 0.01 ^c^	5.28 ± 0.34 ^e^	2.41 ± 0.24 ^e^	0.02 ± 0.01 ^ef^	-	6.92 ± 0.36 ^d^
‘Tianxinbao’	0.09 ± 0.01 ^d^	3.77 ± 0.36 ^h^	3.11 ± 0.29 ^c^	0.03 ± 0.01 ^ef^	0.83 ± 0.12 ^a^	3.23 ± 0.13 ^i^
‘Longcheng No.2’	0.08 ± 0.01 ^d^	7.13 ± 0.44 ^b^	1.25 ± 0.11 ^i^	0.052 ± 0.01 ^bc^	0.47 ± 0.05 ^d^	9.62 ± 0.76 ^c^

Means with different letters in the same column express significant differences (Duncan’s test *p* < 0.05). -: not available.

**Table 3 molecules-28-07559-t003:** OAV analysis of the main aroma compounds of in fruits of different *A. arguta* varieties.

Compound	Threshold (μg/kg)	‘Kuilv’	‘Fenglv’	‘Jialv’	‘Wanlv’	‘Xinlv’	‘Pinglv’	‘Lvbao’	‘Cuiyu’	‘Tianxinbao’	‘Longcheng No.2’
(E)-2-Hexenal	82	84.62	98.44	89.11	72.73	88.68	79.1	27.22	27.27	51.79	54.8
(E)-2-Octenal	4	8.71	9.35	7.14	7.64	10.57	32.91	26.05	29.45	31.85	43.44
(Z)-4-Heptenal	0.8	43.52	48.4	54.9	69.57	66.67	64.24	247.51	79.01	94.39	93.66
1-Hexanal	350	5.87	6.95	6.98	6.68	7.07	4.73	<1	1.3	4.11	3.49
1-Nonanal	1.1	29.29	35.08	27.77	37.71	35.25	31.08	22.53	18.07	25.58	19.66
Valeraldehyde	20	15.95	11.86	16.86	16.04	12.14	8.02	11.28	7.71	10.53	6.98
3-Methyl butanal	80	<1	1.42	1.29	1.95	2.40	2.11	<1	<1	<1	1.24
Benzaldehyde	3	4.82	3.66	3.43	4.20	5.40	4.06	4.25	2.66	4.56	2.78
1-Octen-3-ol	1.5	32.85	57.09	30.91	51.64	69.29	51.72	28.03	23.95	36.51	30.66
1-Penten-3-one	398	<1	1.41	1.07	<1	1.08	<1	<1	<1	<1	<1
Acetoin	55	3.44	1.86	1.93	1.88	1.69	1.78	14.84	24.83	6.64	10.4
α-Phellandrene	40	1.50	<1	<1	6.54	3.52	<1	1.76	<1	1.16	4.92
α-Pinene	2.2	9.55	2.02	2.07	11.45	5.53	3.81	2.18	5.10	4.23	9.51
Myrcene	16.6	1.91	1.18	1.13	8.21	1.17	4.08	1.42	1.63	1.41	1.84
Terpinolene	41	6.14	<1	<1	5.01	2.59	<1	1.17	1.87	<1	4.89
Ethyl butyrate	20	7.37	6.38	3.36	6.68	10.25	4.32	6.57	2.38	3.83	5.66
Butyl isovalerate	78	2.74	2.92	2.4	2.72	2.58	2.26	34.51	8.39	21.18	7.9
Butyl acetate	66	2.45	6.52	4.8	2.52	5.71	7.11	14.75	6.07	4.35	3.66
Hexyl acetate	2	14.08	9.93	10.54	10.66	10.03	11.01	40.50	15.33	19.24	15.57
Hexyl propanoate	8	7.79	3.24	2.46	2.36	2.22	2.05	3.47	5.74	6.83	14.64
Isopentyl acetate	30	4.76	4.29	4.06	4.23	4.47	4.01	100.27	18.62	16.14	8.52
Isobutyl acetate	500	<1	<1	<1	<1	<1	<1	4.15	<1	<1	<1
Methyl isobutyrate	7	19.32	25.68	17.55	18.62	21.68	18.39	12.14	9.38	27.06	18.66
Methyl butanoate	10	2.82	3.72	3.26	6.37	4.37	6.29	1.13	4.91	6.80	6.26

## Data Availability

All related data and methods are presented in this paper. Additional inquiries should be addressed to the corresponding author.

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
