# Peer review of "Flavor Quality Analysis of Ten Actinidia arguta Fruits Based on High-Performance Liquid Chromatography and Headspace Gas Chromatography–Ion Mobility Spectrometry"

_molecules, 2023, doi:10.3390/molecules28227559_

Round 1

Reviewer 1 Report

Comments and Suggestions for Authors

1-The manuscript is about Flavor quality analysis of ten Actinidia arguta fruits based on HPLC and HS-GC-IMS.

2- Please check the statement". Each of the 10 A. arguta varieties had a distinctive 24 flavor. This study can clarify the differences between varieties and provide a reference for the evaluation of arrowroot fruit flavor, variety improvement and new variety selection" Is this correct?

3- it is suggested to develop the table of previous study before methods.

4- It is suggested to illustrate the compounds identified using ChemDraw.

5- It is suggested to include the heading of study strength and study limitation. Study limitation must include all bias in details. at least 3 paragraph.

6- Could you ask your all authors to check your results and illustration on  PCA analysis of A. arguta fruit aroma substances.

7- Please include study validation study and how you make sure for the results to reproduce. It is mandatory to include both in methods and results and explain with detail or include some experiment.

Comments on the Quality of English Language

Okay and publishable.

Author Response

Thank you for your careful checks. We are sorry for our carelessness. Based on the reviewers' and academic editors' comments, we tried our best to improve the manuscript and made some changes. These changes will not affect the content and framework of the paper. Here, we have emphasized these changes in red font in the revised manuscript. We would like to express our heartfelt thanks to all of you for your enthusiastic work and hope that the revised manuscript will be approved by you. See the annex for more details.

Reviewer 2 Report

Comments and Suggestions for Authors

The significance of the manuscript's focus becomes even more pronounced in light of the increasing demand for food. Actinidia arguta fruits have the potential to hold substantial nutritional and economic value, provided that their flavor quality can be enhanced through the careful selection of optimal varieties and the improvement of new variety choices.

The manuscript is clear and rigorous and, the description of the methods and the experimental design is exhaustive and clearly described. The whole presentation is well written, the research design appropriate and all the methods adequately described.

The two following points should be considered to make the manuscript easily readable and more exaustive:

-        (Line 124) a short description (or move the line 172 text) of the “Sugar-acid ratio” will be useful to prevent the reader from having to search for the referenced article.

-        (line 236) “We used the ‘Kuilv’ variety as a reference and subtracted its signal peaks…” Please explain how and why you chose this variety as a reference.

-        Table 7 Please revise the significant figures throughout the table and manuscript: I.e.  “18.76±1.31 “should be written as 19 ± 1 etc. ;  

-        Figure 6 and 10 The quality of the images should be improved to make them legible.

-        Conclusions should be integrated with more details on results summarizing all the work’s results.

In my opinion, the manuscript could be accepted with these minor revisions.

Author Response

Thank you for your careful checks. We are sorry for our carelessness. Based on the reviewers' and academic editors' comments, we tried our best to improve the manuscript and made some changes. These changes will not affect the content and framework of the paper. Here, we have emphasized these changes in red font in the revised manuscript. We would like to express our heartfelt thanks to all of you for your enthusiastic work and hope that the revised manuscript will be approved by you.

Round 2

Reviewer 1 Report

Comments and Suggestions for Authors

You did Detection of Organic Acids Content in this manuscript and similar test was observed in https://www.mdpi.com/2304-8158/12/18/3345. Comprehensive Evaluation of Ten Actinidia arguta Wines Based on Color, Organic Acids, Volatile Compounds, and Quantitative Descriptive Analysis. it is suggested to avoid duplication of experiment. I may be wrong. Please justify. I understand both work is from same authors, but still not recomended. 

Author Response

Thank you for your careful scrutiny. We apologize for our carelessness. Based on the reviewers' and academic editors' comments, we tried our best to improve the manuscript and made some changes. These changes will not affect the content and framework of the paper. Here, we have emphasized these changes with red markers. We would like to express our heartfelt thanks to all of you for your enthusiastic work and hope that the revised manuscript will be approved by all of you.
